# "Short-Dot": Computing Large Linear Transforms Distributedly Using Coded Short Dot Products

**Sanghamitra Dutta**
Carnegie Mellon University
sanghamd@andrew.cmu.edu

**Viveck Cadambe**
Pennsylvania State University
viveck@engr.psu.edu

**Pulkit Grover**
Carnegie Mellon University
pgrover@andrew.cmu.edu

## Abstract

Faced with saturation of Moore's law and increasing size and dimension of data, system designers have increasingly resorted to parallel and distributed computing to reduce computation time of machine-learning algorithms. However, distributed computing is often bottle necked by a small fraction of slow processors called "stragglers" that reduce the speed of computation because the fusion node has to wait for all processors to complete their processing. To combat the effect of stragglers, recent literature proposes introducing redundancy in computations across processors, e.g., using repetition-based strategies or erasure codes. The fusion node can exploit this redundancy by completing the computation using outputs from only a subset of the processors, ignoring the stragglers. In this paper, we propose a novel technique – that we call "Short-Dot" – to introduce redundant computations in a coding theory inspired fashion, for computing linear transforms of long vectors. Instead of computing long dot products as required in the original linear transform, we construct a larger number of redundant and short dot products that can be computed more efficiently at individual processors. Further, only a subset of these short dot products are required at the fusion node to finish the computation successfully. We demonstrate through probabilistic analysis as well as experiments on computing clusters that Short-Dot offers significant speed-up compared to existing techniques. We also derive trade-offs between the length of the dot-products and the resilience to stragglers (number of processors required to finish), for any such strategy and compare it to that achieved by our strategy.

## 1 Introduction

This work proposes a coding-theory inspired computation technique for speeding up computing linear transforms of high-dimensional data by distributing it across multiple processing units that compute shorter dot products. Our main focus is on addressing the "straggler effect," *i.e.*, the problem of delays caused by a few slow processors that bottleneck the entire computation. To address this problem, we provide techniques (building on [1] [2] [3] [4] [5]) that introduce redundancy in the computation by designing a novel error-correction mechanism that allows the size of individual dot products computed at each processor to be shorter than the length of the input.

The problem of computing linear transforms of high-dimensional vectors is "the" critical step [6] in several machine learning and signal processing applications. Dimensionality reduction techniques such as Principal Component Analysis (PCA), Linear Discriminant Analysis (LDA), taking random projections, require the computation of short and fat linear transforms on high-dimensional data. Linear transforms are the building blocks of solutions to various machine learning problems, e.g., regression and classification etc., and are also used in acquiring and pre-processing the data through Fourier transforms, wavelet transforms, filtering, etc. Fast and reliable computation of linear transforms are thus a necessity for low-latency inference [6]. Due to saturation of Moore's law, increasing

speed of computing in a single processor is becoming difficult, forcing practitioners to adopt parallel processing to speed up computing for ever increasing data dimensions and sizes.

Classical approaches of computing linear transforms across parallel processors, e.g., Block-Striped Decomposition [7], Fox's method [8, 7], and Cannon's method [7], rely on dividing the computational task equally among all available processors[1] without any redundant computation. The fusion node collects the outputs from each processors to complete the computation and thus has to wait for all the processors to finish. In almost all distributed systems, a few slow or faulty processors – called "stragglers"[11] – are observed to delay the entire computation. This unpredictable latency in distributed systems is attributed to factors such as network latency, shared resources, maintenance activities, and power limitations. In order to combat with stragglers, cloud computing frameworks like Hadoop [12] employ various straggler detection techniques and usually reset the task allotted to stragglers. Forward error-correction techniques offer an alternative approach to deal with this "straggler effect" by introducing redundancy in the computational tasks across different processors. The fusion node now requires outputs from only a subset of all the processors to successfully finish. In this context, the use of preliminary erasure codes dates back to the ideas of algorithmic fault tolerance [13] [14]. Recently optimized Repetition and Maximum Distance Separable (MDS) [19] codes have been explored [2] [3] [1] [16] to speed up computations.

We consider the problem of computing $\boldsymbol{Ax}$ where $\boldsymbol{A}_{(M \times N)}$ is a given matrix and $\boldsymbol{x}_{(N \times 1)}$ is a vector that is input to the computation ($M \ll N$). In contrast with [1], which also uses codes to compute linear transforms in parallel, we allow the size of individual dot products computed at each processor to be smaller than $N$, the length of the input. Why might one be interested in computing short dot products while performing an overall large linear transform? This is because for distributed digital processors, the computation time is reduced with the number of operations (length of the dot-products). In Sections 4 and 5, we show that the computation speed-up can be increased beyond that obtained in [1]. Another interesting example comes from recent work on designing processing units that exclusively compute dot-products using analog components [17, 18]. These devices are prone to errors and increased delays in convergence when designed for larger dot products.

To summarize, our main contributions are:

1. To compute $\boldsymbol{Ax}$ for a given matrix $\boldsymbol{A}_{(M \times N)}$, we instead compute $\boldsymbol{Fx}$ where we construct $\boldsymbol{F}_{(P \times N)}$ (total no. of processors $P >$ Required no. of dot-products $M$) such that each $N$-length row of $\boldsymbol{F}$ has at most $N(P - K + M)/P$ non-zero elements. Because the locations of zeros in a row of $\boldsymbol{F}$ are known by design, this reduces the complexity of computing dot-products of rows of $\boldsymbol{F}$ with $\boldsymbol{x}$. Here $K$ parameterizes the resilience to stragglers: any $K$ of the $P$ dot products of rows of $\boldsymbol{F}$ with $\boldsymbol{x}$ are sufficient to recover $\boldsymbol{Ax}$, *i.e.*, any $K$ rows of $\boldsymbol{F}$ can be linearly combined to generate the rows of $\boldsymbol{A}$.
2. We provide fundamental limits on the trade-off between the length of the dot-products and the straggler resilience (number of processors to wait for) for *any* such strategy in Section 3. This suggests a lower bound on the length of task allotted per processor. However, we believe that these limits are loose and point to an interesting direction for future work.
3. Assuming exponential tails of service-times at each server (used in [1]), we derive the expected computation time required by our strategy and compare it to uncoded parallel processing, repetition strategy and MDS codes [19] (see Fig. 2). Short-Dot offers speed-up by a factor of $\Omega(\log(P))$ over uncoded, parallel processing and repetition, and nearly by a factor of $\Omega(\frac{P}{M})$ compared to MDS codes when $M$ is linear in $P$. The strategy out-performs repetition or MDS codes by a factor of $\Omega\left(\frac{P}{M \log(P/M)}\right)$ when $M$ is sub-linear in $P$.
4. We provide experimental results showing that Short-Dot is faster than existing strategies.

For the rest of the paper, we define the sparsity of a vector $\boldsymbol{u} \in \mathbb{R}^N$ as the number of nonzero elements in the vector, *i.e.*, $\|\boldsymbol{u}\|_0 = \sum_{j=1}^{N} \mathcal{I}(u_j \neq 0)$. We also assume that $P$ divides $N$ ($P \ll N$).

**Comparison with existing strategies:** Consider the problem of computing a single dot product of an input vector $\boldsymbol{x} \in \mathbb{R}^N$ with a pre-specified vector $\boldsymbol{a} \in \mathbb{R}^N$. By an "uncoded" parallel processing strategy (which includes Block Striped Decomposition [7]), we mean a strategy that does not use redundancy to overcome delays caused by stragglers. One uncoded strategy is to partition the dot product into $P$ smaller dot products, where $P$ is the number of available processors. E.g. $\boldsymbol{a}$ can

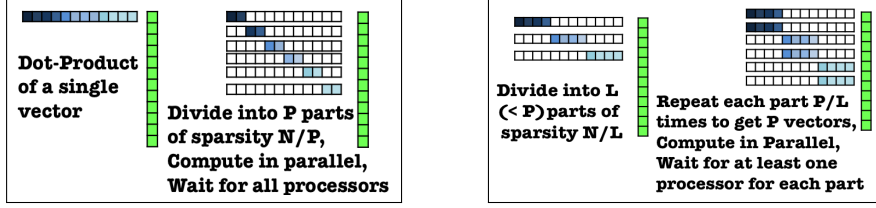

Figure 1: A dot-product of length $N = 12$ is being computed parallely using $P = 6$ processors. (Left) Uncoded Parallel Processing - Divide into $P$ parts, (Right) Repetition with block partitioning.

be divided into $P$ parts – constructing $P$ short vectors of sparsity $N/P$ – with each vector stored in a different processor (as shown in Fig. 1 left). Only the nonzero values of the vector need to be stored since the locations of the nonzero values is known apriori at every node. One might expect the computation time for each processor to reduce by a factor of $P$. However, now the fusion node has to wait for all the $P$ processors to finish their computation, and the stragglers can now delay the entire computation. Can we construct $P$ vectors such that dot products of a subset of them with $\boldsymbol{x}$ are sufficient to compute $\langle \boldsymbol{a}, \boldsymbol{x} \rangle$? A simple coded strategy is Repetition with block partitioning *i.e.*, constructing $L$ vectors of sparsity $N/L$ by partitioning the vector of length $N$ into $L$ parts ($L < P$), and repeating the $L$ vectors $P/L$ times so as to obtain $P$ vectors of sparsity $N/L$ as shown in Fig. 1 (right). For each of the $L$ parts of the vector, the fusion node only needs the output of one processor among all its repetitions. Instead of a single dot-product, if one requires the dot-product of $\boldsymbol{x}$ with $M$ vectors $\{\boldsymbol{a}_1, \ldots, \boldsymbol{a}_M\}$, one can simply repeat the aforementioned strategy $M$ times.

For multiple dot-products, an alternative repetition-based strategy is to compute $M$ dot products $P/M$ times in parallel at different processors. Now we only have to wait for at least one processor corresponding to each of the $M$ vectors to finish. Improving upon repetition, it is shown in [1] that an $(P, M)$-MDS code allows constructing $P$ coded vectors such that any $M$ of $P$ dot-products can be used to reconstruct all the $M$ original vectors (see Fig. 2b). This strategy is shown, both experimentally and theoretically, to perform better than repetition and uncoded strategies.

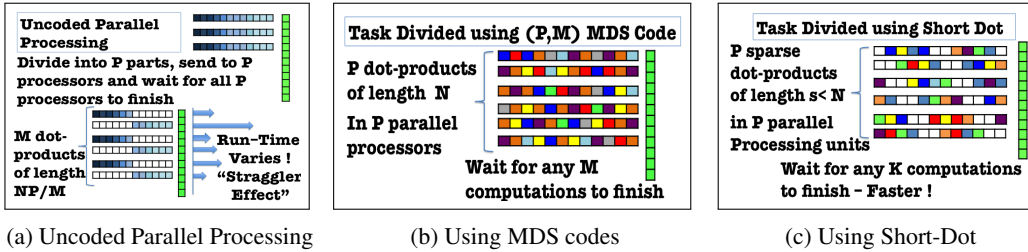

(a) Uncoded Parallel Processing     (b) Using MDS codes     (c) Using Short-Dot

Figure 2: Different strategies of parallel processing: Here $M = 3$ dot-products of length $N = 12$ are being computed using $P = 6$ processors.

*Can we go beyond MDS codes?* MDS codes-based strategies require $N$-length dot-products to be computed on each processor. Short-Dot instead constructs $P$ vectors of sparsity $s$ (less than $N$), such that the dot product of $\boldsymbol{x}$ with any $K$ ($\geq M$) out of these $P$ short vectors is sufficient to compute the dot-product of $\boldsymbol{x}$ with all the $M$ given vectors (see Fig. 2c). Compared to MDS Codes, Short-Dot waits for some more processors (since $K \geq M$), but each processor computes a shorter dot product. We also propose Short-MDS, an extension of the MDS codes-based strategy in [1] to create short dot-products of length $s$, through block partitioning, and compare it with Short-Dot. In regimes where $\frac{N}{s}$ is an integer, Short-MDS may be viewed as a special case of Short-Dot. But when $\frac{N}{s}$ is not an integer, Short-MDS has to wait for more processors in worst case than Short-Dot for the same sparsity $s$, as discussed in Remark 1 in Section 2.

## 2    Our coded parallelization strategy: Short-Dot

In this section, we provide our strategy of computing the linear transform $\boldsymbol{Ax}$ where $\boldsymbol{x} \in \mathbb{R}^N$ is the input vector and $\boldsymbol{A}_{(M \times N)} = [\boldsymbol{a}_1, \boldsymbol{a}_2, \ldots, \boldsymbol{a}_M]^T$ is a given matrix. Short-Dot constructs a

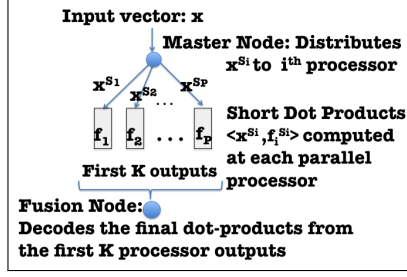

Figure 3: Short-Dot: Distributes short dot-products over $P$ parallel processors, such that outputs from any $K$ out of $P$ processors are sufficient to compute successfully.

$P \times N$ matrix $\boldsymbol{F} = [\boldsymbol{f}_1, \boldsymbol{f}_2, \ldots, \boldsymbol{f}_P]^T$ such that $M$ predetermined linear combinations of *any* $K$ rows of $\boldsymbol{F}$ are sufficient to generate each of $\{\boldsymbol{a}_1^T, \ldots, \boldsymbol{a}_M^T\}$, and any row of $\boldsymbol{F}$ has sparsity at most $s = \frac{N}{P}(P - K + M)$. Each sparse row of $\boldsymbol{F}$ (say $\boldsymbol{f}_i^T$) is sent to the $i$-th processor ($i = 1, \ldots, P$) and dot-products of $\boldsymbol{x}$ with all sparse rows are computed in parallel. Let $S_i$ denote the support (set of non-zero indices) of $\boldsymbol{f}_i$. Thus, for any unknown vector $\boldsymbol{x}$, short dot products of length $|S_i| \leq s = \frac{N}{P}(P - K + M)$ are computed on each processor. Since the linear combination of any $K$ rows of $\boldsymbol{F}$ can generate the rows of $\boldsymbol{A}$, i.e., $\{\boldsymbol{a}_1^T, \boldsymbol{a}_2^T, \ldots, \boldsymbol{a}_M^T\}$, the dot-product from the earliest $K$ out of $P$ processors can be linearly combined to obtain the linear transform $\boldsymbol{A}\boldsymbol{x}$. Before formally stating our algorithm, we first provide an insight into why such a matrix $\boldsymbol{F}$ exists in the following theorem, and develop an intuition on the construction strategy.

**Theorem 1** *Given row vectors $\{\boldsymbol{a}_1^T, \boldsymbol{a}_2^T, \ldots, \boldsymbol{a}_M^T\}$, there exists a $P \times N$ matrix $\boldsymbol{F}$ such that a linear combination of any $K(> M)$ rows of the matrix is sufficient to generate the row vectors and each row of $\boldsymbol{F}$ has sparsity at most $s = \frac{N}{P}(P - K + M)$, provided $P$ divides $N$.*

*Proof:* We may append $(K - M)$ rows to $\boldsymbol{A} = [\boldsymbol{a}_1, \boldsymbol{a}_2, \ldots, \boldsymbol{a}_M]^T$, to form a $K \times N$ matrix $\tilde{\boldsymbol{A}} = [\boldsymbol{a}_1, \boldsymbol{a}_2, \ldots, \boldsymbol{a}_M, \boldsymbol{z}_1, \ldots, \boldsymbol{z}_{K-M}]^T$. The precise choice of these additional vectors will be made explicit later. Next, we choose $\boldsymbol{B}$, a $P \times K$ matrix such that *any square sub-matrix of $\boldsymbol{B}$ is invertible*. E.g., A Vandermonde or Cauchy Matrix, or a matrix with *i.i.d.* Gaussian entries can be shown to satisfy this property with probability 1. The following lemma shows that *any $K$ rows of the matrix $\boldsymbol{B}\tilde{\boldsymbol{A}}$ are sufficient to generate any row of $\tilde{\boldsymbol{A}}$, including $\{\boldsymbol{a}_1^T, \boldsymbol{a}_2^T, \ldots, \boldsymbol{a}_M^T\}$:*

**Lemma 1** *Let $\boldsymbol{F} = \boldsymbol{B}\tilde{\boldsymbol{A}}$ where $\tilde{\boldsymbol{A}}$ is a $K \times N$ matrix and $\boldsymbol{B}$ is any $(P \times K)$ matrix such that every square sub-matrix is invertible. Then, any $K$ rows of $\boldsymbol{F}$ can be linearly combined to generate any row of $\tilde{\boldsymbol{A}}$.*

*Proof:* Choose an arbitrary index set $\chi \subset \{1, 2, \ldots, P\}$ such that $|\chi| = K$. Let $\boldsymbol{F}^\chi$ be the sub-matrix formed by chosen $K$ rows of $\boldsymbol{F}$ indexed by $\chi$. Then, $\boldsymbol{F}^\chi = \boldsymbol{B}^\chi \tilde{\boldsymbol{A}}$. Now, $\boldsymbol{B}^\chi$ is a $K \times K$ sub-matrix of $\boldsymbol{B}$, and is thus invertible. Thus, $\tilde{\boldsymbol{A}} = (\boldsymbol{B}^\chi)^{-1} \boldsymbol{F}^\chi$. The $i$-th row of $\tilde{\boldsymbol{A}}$ is $[i$-th Row of $(\boldsymbol{B}^\chi)^{-1}]\boldsymbol{F}^\chi$ for $i = 1, 2, \ldots, K$. Thus, each row of $\tilde{\boldsymbol{A}}$ is generated by the chosen $K$ rows of $\boldsymbol{F}$. ∎

In the next lemma, we show how the row sparsity of $\boldsymbol{F}$ can be constrained to be at most $\frac{N}{P}(P - K + M)$ by appropriately choosing the appended vectors $\boldsymbol{z}_1, \ldots, \boldsymbol{z}_{K-M}$.

**Lemma 2** *Given an $M \times N$ matrix $\boldsymbol{A} = [\boldsymbol{a}_1, \ldots, \boldsymbol{a}_M]^T$, let $\tilde{\boldsymbol{A}} = [\boldsymbol{a}_1, \ldots, \boldsymbol{a}_M, \boldsymbol{z}_1, \ldots, \boldsymbol{z}_{K-M}]^T$ be a $K \times N$ matrix formed by appending $K - M$ row vectors to $\boldsymbol{A}$. Also let $\boldsymbol{B}$ be a $P \times K$ matrix such that every square matrix is invertible. Then there exists a choice of the appended vectors $\boldsymbol{z}_1, \ldots, \boldsymbol{z}_{K-M}$ such that each row of $\boldsymbol{F} = \boldsymbol{B}\tilde{\boldsymbol{A}}$ has sparsity at most $s = \frac{N}{P}(P - K + M)$.*

*Proof:* We select a sparsity pattern that we want to enforce on $\boldsymbol{F}$ and then show that there exists a choice of the appended vectors $\boldsymbol{z}_1, \ldots, \boldsymbol{z}_{K-M}$ such that the pattern can be enforced.
**Sparsity Pattern enforced on $\boldsymbol{F}$:** This is illustrated in Fig. 4. First, we construct a $P \times P$ "unit block" with a cyclic structure of nonzero entries, where $(K - M)$ zeros in each row and column are arranged as shown in Fig. 4. Each row and column have at most $s_c = P - K + M$ non-zero entries. This unit block is replicated horizontally $N/P$ times to form an $P \times N$ matrix with at most

$s_c$ non-zero entries in each column, and and at most $s = Ns_r/P$ non-zero entries in each row. We now show how choice of $\boldsymbol{z}_1, \ldots, \boldsymbol{z}_{K-M}$ can enforce this pattern on $\boldsymbol{F}$.

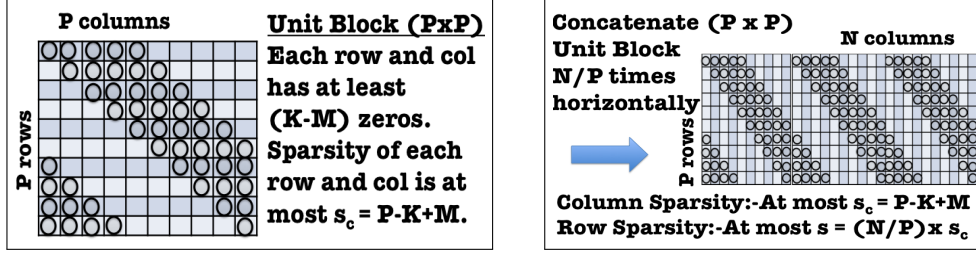

Figure 4: Sparsity pattern of $\boldsymbol{F}$: (Left) Unit Block ($P \times P$); (Right) Unit Block concatenated $N/P$ times to form $N \times P$ matrix $\boldsymbol{F}$ with row sparsity at most $s$.

From $\boldsymbol{F} = \boldsymbol{B}\tilde{\boldsymbol{A}}$, the $j$-th column of $\boldsymbol{F}$ can be written as, $\boldsymbol{F}_j = \boldsymbol{B}\tilde{\boldsymbol{A}}_j$. Each column of $\boldsymbol{F}$ has at least $K - M$ zeros at locations indexed by $U \subset \{1, 2, \ldots, P\}$. Let $\boldsymbol{B}^U$ denote a $((K - M) \times K)$ sub-matrix of $\boldsymbol{B}$ consisting of the rows of $\boldsymbol{B}$ indexed by $U$. Thus, $\boldsymbol{B}^U \tilde{\boldsymbol{A}}_j = [\boldsymbol{0}]_{(K-M) \times 1}$. Divide $\tilde{\boldsymbol{A}}_j$ into two portions of lengths $M$ and $K - M$ as follows:
$\tilde{\boldsymbol{A}}_j = [\boldsymbol{A}_j^T \mid \boldsymbol{z}^T]^T = [a_1(j)\, a_2(j) \ldots a_M(j)\ \ z_1(j)\ \ldots\ z_{K-M}(j)]^T$
Here $\boldsymbol{A}_j = [a_1(j)\, a_2(j) \ldots a_M(j)]^T$ is actually the $j$-th column of given matrix $\boldsymbol{A}$ and $\boldsymbol{z} = [z_1(j), \ldots z_{K-M}(j)]^T$ depends on the choice of the appended vectors. Thus,

$$\boldsymbol{B}_{cols\ 1:M}^U \boldsymbol{A}_j + \boldsymbol{B}_{cols\ M+1:K}^U \boldsymbol{z} = [\boldsymbol{0}]_{K-M \times 1} \qquad \Rightarrow \boldsymbol{B}_{cols\ M+1:K}^U \boldsymbol{z} = -\boldsymbol{B}_{cols\ 1:M}^U [\boldsymbol{A}_j]$$
$$\Rightarrow [\,\boldsymbol{z}\,] = -(\boldsymbol{B}_{cols\ M+1:K}^U)^{-1} \boldsymbol{B}_{cols\ 1:M}^U [\boldsymbol{A}_j] \tag{1}$$

where the last step uses the fact that $[\boldsymbol{B}_{cols\ M+1:K}^U]$ is invertible because it is a $(K - M) \times (K - M)$ square sub-matrix of $\boldsymbol{B}$. This explicitly provides the vector $\boldsymbol{z}$ which completes the $j$-th column of $\tilde{\boldsymbol{A}}$. The other columns of $\tilde{\boldsymbol{A}}$ can be completed similarly, proving the lemma. ∎

From Lemmas 1 and 2, for a given $M \times N$ matrix $\boldsymbol{A}$, there always exists a $P \times N$ matrix $\boldsymbol{F}$ such that a linear combination of *any* $K$ columns of $\boldsymbol{F}$ is sufficient to generate our given vectors and each row of $\boldsymbol{F}$ has sparsity at most $s = \frac{N}{P}(P - K + M)$. This proves the theorem. ∎

With this insight in mind, we now formally state our computation strategy:

---

**Algorithm 1** Short-Dot

---

**[A] Pre-Processing Step: Encode $\boldsymbol{F}$ (Performed Offline)**
**Given:** $\boldsymbol{A}_{M \times N} = [\boldsymbol{a}_1, \ldots, \boldsymbol{a}_M]^T = [\boldsymbol{A}_1, \boldsymbol{A}_2, \ldots, \boldsymbol{A}_N],\ parameter\ K, Matrix\ \boldsymbol{B}_{P \times K}$
1: **For** $j = 1\ to\ N$ **do**
2:     **Set**   $U \leftarrow (\{(j-1), \ldots, (j + K - M - 1)\}\ \mod P) + 1$
3:                       $\triangleright$ The set of $(K - M)$ indices that are 0 for the $j$-th column of $\boldsymbol{F}$
4:     **Set**   $\boldsymbol{B}^U \leftarrow$ Rows of $\boldsymbol{B}$ indexed by $U$
5:     **Set**   $[\,\boldsymbol{z}\,] = -(\boldsymbol{B}_{cols\ M+1:K}^U)^{-1} \boldsymbol{B}_{cols\ 1:M}^U [\boldsymbol{A}_j]$       $\triangleright$ $\boldsymbol{z}_{(K-M) \times 1}$ is a row vector.
6:     **Set**   $\boldsymbol{F}_j = \boldsymbol{B}[\boldsymbol{A}_j^T | \boldsymbol{z}^T]^T$         $\triangleright$ $\boldsymbol{F}_j$ is a column vector ( $j$-th col of $\boldsymbol{F}$)
    **Encoded Output:** $\boldsymbol{F}_{P \times N} = [\boldsymbol{f}_1 \boldsymbol{f}_2 \ldots \boldsymbol{f}_P]^T$             $\triangleright$ Row representation of matrix $\boldsymbol{F}$
7: **For** $i = 1\ to\ P$ **do**
8:     **Store**   $S_i \leftarrow Support(\boldsymbol{f}_i)$         $\triangleright$ Indices of non-zero entries in the $i$-th row of $\boldsymbol{F}$
9:     **Send**   $\boldsymbol{f}_i^{S_i}$ to $i$-th processor              $\triangleright$ $i$-th row of $\boldsymbol{F}$ sent to $i$-th processor
**[B] Online computations**
**External Input :** $\boldsymbol{x}$
**Resources:** $P$ parallel processors $(P > M)$
**[B1] Parallelization Strategy: Divide task among parallel processors:**
1: **For** $i = 1\ to\ P$ **do**
2:     Send $\boldsymbol{x}^{S_i}$ to the $i$-th processor
3:     Compute at $i$-th processor: $\langle \boldsymbol{f}_i^{S_i}, \boldsymbol{x}^{S_i} \rangle \triangleright \boldsymbol{u}^S$ denotes only the rows of vector $\boldsymbol{u}$ indexed by $S$
    **Output:** $\langle \boldsymbol{f}_i^{S_i}, \boldsymbol{x}^{S_i} \rangle$ from $K$ earliest processors

---

---

**[B2] Fusion Node: Decode the dot-products from the processor outputs:**
1: **Set** $V \leftarrow$ Indices of the K processors that finished first
2: **Set** $\boldsymbol{B}^V \leftarrow$ Rows of $\boldsymbol{B}$ indexed by $V$
3: **Set** $\boldsymbol{v}_{K \times 1} \leftarrow [\langle \boldsymbol{f}_i^{S_i}, \boldsymbol{x}^{S_i} \rangle, \, \forall\, i \in V]$      $\triangleright$ Col Vector of outputs from first $K$ processors
4: **Set** $\boldsymbol{Ax} = [\langle \boldsymbol{a}_1, \boldsymbol{x} \rangle, \dots, \langle \boldsymbol{a}_M, \boldsymbol{x} \rangle]^T \leftarrow [(\boldsymbol{B}^V)^{-1}]^{rows\ 1:M} \boldsymbol{v}$
5: **Output:** $\langle \boldsymbol{x}, \boldsymbol{a}_1 \rangle, \dots, \langle \boldsymbol{x}, \boldsymbol{a}_M \rangle$

---

Table 1: Trade-off between the length of the dot-products and parameter $K$ for different strategies

| Strategy | Length | Parameter $K$ | Strategy | Length | Parameter $K$ |
|---|---|---|---|---|---|
| Repetition | $N$ | $P - \lfloor \frac{P}{M} \rfloor + 1$ | Repetition with block partition | $s$ | $P - \lfloor \frac{P}{M \lceil N/s \rceil} \rfloor + 1$ |
| MDS | $N$ | $M$ | | | |
| Short-Dot | $s$ | $P - \lfloor \frac{Ps}{N} \rfloor + M$ | Short-MDS | $s$ | $P - \lfloor \frac{P}{\lceil N/s \rceil} \rfloor + M$ |

**Remark 1: Short-MDS - a special case of Short-Dot** An extension of the MDS codes-based strategy proposed in [1], that we call Short-MDS can be designed to achieve row-sparsity $s$. First block-partition the matrix of $N$ columns, into $\lceil N/s \rceil$ sub-matrices of size $M \times s$, and also divide the total processors $P$ equally into $\lceil N/s \rceil$ parts. Now, each sub-matrix can be encoded using a $(\frac{P}{\lceil N/s \rceil}, M)$ MDS code. In the worst case, including all integer effects, this strategy requires $K = P - \lfloor \frac{P}{\lceil N/s \rceil} \rfloor + M$ processors to finish. In comparison, Short-Dot requires $K = P - \lfloor \frac{Ps}{N} \rfloor + M$ processors to finish. In the regime where, $s$ exactly divides $N$, Short-MDS can be viewed as a special case of Short-Dot, as both the expressions match. However, in the regime where $s$ does not exactly divide $N$, Short-MDS requires more processors to finish in the worst case than Short-Dot. Short-Dot is a generalized framework that can achieve a wider variety of pre-specified sparsity patterns as required by the application. In Table 1, we compare the lengths of the dot-products and straggler resilience $K$, *i.e.*, the number of processors to wait for in worst case, for different strategies.

## 3    Limits on trade-off between the length of dot-products and parameter *K*

**Theorem 2** *Let $\boldsymbol{A}_{M \times N}$ be any matrix such that each column has at least one non-zero element. If the linear combination of any $K$ rows of $\boldsymbol{F}_{(P \times N)}$ can generate $M$ rows of $\boldsymbol{A}_{M \times N}$, then the average sparsity $s$ of each row of $\boldsymbol{F}_{(P \times N)}$ must satisfy $s \geq N \left(1 - \frac{K}{P}\right) + \frac{N}{P}$.*

*Proof:* We claim that $K$ is strictly greater than the maximum number of zeros that can occur in any column of the matrix $\boldsymbol{F}$. If not, suppose the $j$-th column of $\boldsymbol{F}$ has more than $K$ zeros. Then there exists a linear combination of $K$ rows of $\boldsymbol{F}$ that will always have $0$ at the $j$-th column index and it is not possible to generate any row of the given matrix $\boldsymbol{A}$. Thus, $K$ is no less than $1 + Max\ No.\ of\ 0s\ in\ any\ column\ of\ \boldsymbol{F}$. Since, maximum value is always greater than average,

$$K \geq 1 + Avg.\ No.\ of\ 0s\ in\ any\ column\ of\ \boldsymbol{F} \geq 1 + \frac{(N-s)P}{N}. \tag{2}$$

A slight re-arrangement establishes the aforementioned lower bound.    ∎
Short-Dot achieves a row-sparsity of at most $s = N \left(1 - \frac{K}{P}\right) + \frac{NM}{P}$ while the lower bound for any such strategy is $s \geq N \left(1 - \frac{K}{P}\right) + \frac{N}{P}$. Notice that the bounds only differ in the second term. We believe that the difference in the bounds arises due to the looseness of the fundamental limit: our technique is based on derivation for $M = 1$ (bound is tight), and could be tightened for $M > 1$.

## 4    Analysis of expected computation time for exponential tail models

We now provide a probabilistic analysis of the computational time required by Short-Dot and compare it with uncoded parallel processing, repetition and MDS codes as shown in Fig. 5. Table 2 shows the order-sense expected computation time in the regimes where $M$ is linear and sub-linear in $P$. A detailed analysis is provided in the supplement. Assume that the time required by a processor to

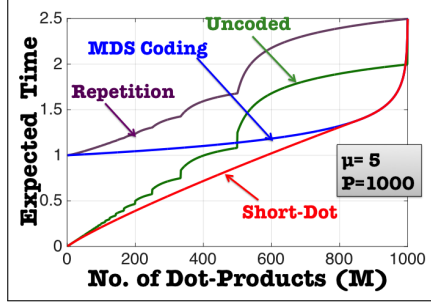

Figure 5: Expected computation time: Short-Dot is faster than MDS when $M \ll P$ and Uncoded when $M \approx P$, and is universally faster over the entire range of $M$. For the choice of straggling parameter, Repetition is slowest. When $M$ does not exactly divide $P$, the distribution of computation time for repetition and uncoded strategies is the maximum of non-identical but independent random variables, which produce the ripples in these curves (see supplement for details).

compute a single dot-product follows an exponential distribution and is independent of the other processors, as described in [1]. Let the time required to compute a single dot-product of length $N$ be distributed as: $\Pr(T_N \leq t) = 1 - \exp\left(-\mu\left(\frac{t}{N} - 1\right)\right) \ \forall \ t \geq N$. Here, $\mu$ is the "straggling parameter" that determines the unpredictable latency in computation time. For an $s$ length dot product, we simply replace $N$ by $s$. The expected computation time for Short-Dot is the expected value of the $K$-th order statistic of these $P$ *iid* exponential random variables, which is given by:

$$E(T) = s\left(1 + \frac{\log(\frac{P}{P-K})}{\mu}\right) = \frac{(P-K+M)N}{P}\left(1 + \frac{\log(\frac{P}{P-K})}{\mu}\right). \qquad (3)$$

Here, (3) uses the fact that the expected value of the $K$-th statistic of $P$ *iid* exponential random variables with parameter 1 is $\sum_{i=1}^{P}\frac{1}{i} - \sum_{i=1}^{P-K}\frac{1}{i} \approx \log(P) - \log(P-K)$ [1]. The expected computation time in the RHS of (3) is minimized when $P - K = \Theta(M)$. This minimal expected time is $\mathcal{O}(\frac{MN}{P})$ for $M$ linear in $P$ and is $\mathcal{O}\left(\frac{MN\log(P/M)}{P}\right)$ for $M$ sub-linear in $P$.

Table 2: Probabilistic Computation Times

| Strategy | $E(T)$ | $M$ linear in $P$ | $M$ sub-linear in $P$ |
|---|---|---|---|
| Only one Processor | $MN\left(1 + \frac{1}{\mu}\right)$ | $\Theta\left(MN\right)$ | $\Theta\left(MN\right)$ |
| Uncoded (M divides P)[2] | $\frac{MN}{P}\left(1 + \frac{\log(P)}{\mu}\right)$ | $\Theta\left(\frac{MN}{P}\log(P)\right)$ | $\Theta\left(\frac{MN}{P}\log(P)\right)$ |
| Repetition (M divides P) [2] | $N\left(1 + \frac{M\log(M)}{P\mu}\right)$ | $\Theta\left(\frac{MN}{P}\log(P)\right)$ | $\Theta\left(N\right)$ |
| MDS | $N\left(1 + \frac{\log\left(\frac{P}{P-M}\right)}{\mu}\right)$ | $\Theta(N)$ | $\Theta(N)$ |
| Short-Dot | $\frac{N(P-K+M)}{P}\left(1 + \frac{\log\left(\frac{P}{P-K}\right)}{\mu}\right)$ | $\mathcal{O}(\frac{MN}{P})$ | $\mathcal{O}\left(\frac{MN}{P}\log\left(\frac{P}{M}\right)\right)$ |

[2] Refer to Supplement for more accurate analysis taking integer effects into account

**Encoding and Decoding Complexity**: Even though encoding is a pre-processing step (since $\boldsymbol{A}$ is assumed to be given in advance), we include a complexity analysis for the sake of completeness. The encoding requires $\frac{N}{P}$ matrix inversions of size $(K - M)$, and a $P \times K$ matrix multiplication with a $K \times N$ matrix. The naive encoding complexity is therefore $\mathcal{O}(\frac{N}{P}(K-M)^3 + NKP)$. This is higher than MDS codes that has an encoding complexity of $\mathcal{O}(NMP))$, but it is only a one-time cost that provides savings in online steps (as discussed earlier in this section). The decoding complexity of Short-Dot is $\mathcal{O}(K^3 + KM)$ which does not depend on $N$ when $M, K \ll N$. This is nearly the same as $\mathcal{O}(M^3 + M^2)$ complexity of MDS codes. We believe that the complexities might be reduced further, based on special choices of encoding matrix $\boldsymbol{B}$.

Table 3: Experimental computation time of 10000 dot products ($N = 785, M = 10, P = 20$)

| Strategy | Parameter $K$ | Mean | STDEV | Minimum Time | Maximum Time |
|---|---|---|---|---|---|
| Uncoded | 20 | 11.8653 | 2.8427 | 9.5192 | 27.0818 |
| Short-Dot | 18 | 10.4306 | 0.9253 | 8.2145 | 11.8340 |
| MDS | 10 | 15.3411 | 0.8987 | 13.8232 | 17.5416 |

## 5  Experimental Results

We perform experiments on computing clusters at CMU to test the computational time. We use HTCondor [20] to schedule jobs simultaneously among the $P$ processors. We compare the time required to classify 10000 handwritten digits of the MNIST [21] database, assuming we are given a trained 1-layer Neural Network. We separately trained the Neural network using training samples, to form a matrix of weights, denoted by $A_{10 \times 785}$. For testing, the multiplication of this given $10 \times 785$ matrix, with the test data matrix $X_{785 \times 10000}$ is considered. The total number of processors was 20.

Assuming that $A_{10 \times 785}$ is encoded into $F_{20 \times 785}$ in a pre-processing step, we store the rows of $F$ in each processor apriori. Now portions of the data matrix $X$ of size $s \times 10000$ are sent to each of the $P$ parallel processors as input. We also send a C-program to compute dot-products of length $s = \frac{N}{P}(P - K + M)$ with appropriate rows of $F$ using command *condor-submit*. Each processor outputs the value of one dot-product. The computation time reported in Fig. 6 includes the total time required to communicate inputs to each processor, compute the dot-products in parallel, fetch the required outputs, decode and classify all the 10000 test-images, based on 35 experimental runs.

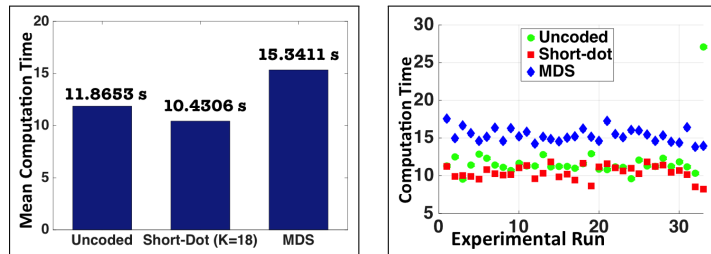

Figure 6: Experimental results: (Left) Mean computation time for Uncoded Strategy, Short-Dot (K=18) and MDS codes: Short-Dot is faster than MDS by 32% and Uncoded by 12%. (Right) Scatter plot of computation time for different experimental runs: Short-Dot is faster most of the time.

**Key Observations:** (See Table 3 for detailed results). Computation time varies based on nature of straggling, at the particular instant of the experimental run. Short-Dot outperforms both MDS and Uncoded, in mean computation time. Uncoded is faster than MDS since per-processor computation time for MDS is larger, and it increases the straggling, even though MDS waits for only for 10 out of 20 processors. However, note that Uncoded has more variability than both MDS and Short-Dot, and its maximum time observed during the experiment is much greater than both MDS and Short-Dot. The classification accuracy was 85.98% on test data.

## 6  Discussion

While we have presented the case of $M < P$ here, Short-Dot easily generalizes to the case where $M \geq P$. The matrix can be divided horizontally into several chunks along the row dimension (shorter matrices) and Short-Dot can be applied on each of those chunks one after another. Moreover if rows with same sparsity pattern are grouped together and stored in the same processor initially, then the communication cost is also significantly reduced during the online computations, since only some elements of the unknown vector $x$ are sent to a particular processor.

**Acknowledgments:** Systems on Nanoscale Information fabriCs (SONIC), one of the six SRC STARnet Centers, sponsored by MARCO and DARPA. We also acknowledge NSF Awards 1350314, 1464336 and 1553248. S Dutta also received Prabhu and Poonam Goel Graduate Fellowship.

## Footnotes

[1]Strassen's algorithm [9] and its generalizations offer a recursive approach to faster matrix multiplications over multiple processors, but they are often not preferred because of their high communication cost [10].

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
