[Supplementary Material]

# "Short-Dot": Computing Large Linear Transforms Distributedly Using Coded Short Dot Products Supplement

**Sanghamitra Dutta**
Carnegie Mellon University
sanghamd@andrew.cmu.edu

**Viveck Cadambe**
Pennsylvania State University
viveck@engr.psu.edu

**Pulkit Grover**
Carnegie Mellon University
pgrover@andrew.cmu.edu

## 1 Analysis of expected computation time for exponential tail models

We now provide a probabilistic analysis of the computational time required by Short-Dot and compare it with uncoded parallel processing, repetition and MDS codes as shown in Fig. 1.

Figure 1: Comparison of theoretical computation time: Short-Dot outperforms MDS Codes when $M \ll P$ and Uncoded when $M \approx P$, and is universally faster over the entire range of $M$. For the choice of straggling parameters, repetition performs worse than all other strategies.

We assume that the time required by a processor to compute a single dot-product follows an exponential distribution and is independent of other parallel processors.

Let us assume, the time required to compute a single dot-product of length $N$, follow the distribution:-

$$\Pr(T_N \leq t) = F(t) = 1 - \exp\left(-\mu\left(\frac{t}{N} - 1\right)\right) \quad \forall \, t \geq N \tag{1}$$

Here, $\mu$ is a straggling parameter, that determines the "unpredictable latency" in computation time. We also assume, that if the length of the dot-product reduces by a factor of $\tau$, *i.e.*, if the length of the dot-product to be computed changes to $N/\tau$ from $N$, the probability distribution of the computational time varies as:-

$$\Pr(T \leq t) = F(\tau t) = 1 - \exp\left(-\mu\left(\frac{\tau t}{N} - 1\right)\right) \quad \forall \, t \geq N/\tau \tag{2}$$

Thus, if length of the dot-product is $s$ where $s$ is the sparsity of the vector, the computation time would follow the distribution $F(\frac{Nt}{s})$. Now we derive the expected computation time using our proposed strategy and compare it with existing strategies in the regimes where the number of dot-products $M$ is linear and sub-linear in $P$.

Table 1 shows the order-sense expected computation time in the regimes where $M$ is linear and sub-linear in $P$.

## 1.1 Proposed Strategy – Short-Dot:

The computation time over each of the $P$ processors behaves as independent, identically distributed exponential random variables following the distribution:-

$$\Pr(T \le t) = F\left(\frac{Nt}{s}\right) = 1 - \exp\left(-\mu\left(\frac{t}{s} - 1\right)\right) \quad \forall\, t \ge s \tag{3}$$

Now, the expected computation time is the expected value of the $K$-th order statistic of these $P$ independent, identically distributed exponential random variables, which is given by:-

$$E(T) = s\left(1 + \frac{\log(\frac{P}{P-K})}{\mu}\right) = \frac{(P-K+M)N}{P}\left(1 + \frac{\log(\frac{P}{P-K})}{\mu}\right) \tag{4}$$

Here we use the result that the $K-$ th order statistic of $P$ exponential random variables that are independent and identically distributed as $\sim \exp(-T) \,\forall\, T \ge 0$ is given by $\sum_{i=1}^{P} \frac{1}{i} - \sum_{i=1}^{P-K} \frac{1}{i}$. For large $P$ and $K < P$, this can be approximated as $\log(P) - \log(P-K)$.

Note that, the expected computation time is minimized when $P - K = \Theta(M)$, and is given by:-

$$E(T) = \mathcal{O}\left(\frac{MN}{P}\left(1 + \frac{\log(P/M)}{\mu}\right)\right) \tag{5}$$

If $M$ is linear in $P$, the expected time is $\mathcal{O}(\frac{MN}{P})$. If $M$ is sub-linear in $P$, the expected time is $\mathcal{O}\left(\frac{MN\log(P/M)}{P}\right)$. Note that, $s = \frac{(P-K+M)N}{P}$ is actually an upper bound on the length of each dot-product, using Short-Dot. Thus the expression obtained in (5) is an upper bound for the actual computation time. Thus we use $\mathcal{O}(.)$ instead of $\Theta(.)$.

Table 1: Probabilistic Computation Times

| Method | $E(T)$ | $M$ linear in $P$ | $M$ sub-linear in $P$ |
|---|---|---|---|
| Only one Processor | $MN\left(1 + \frac{1}{\mu}\right)$ | $\Theta\left(MN\right)$ | $\Theta\left(MN\right)$ |
| Uncoded [1] | $\frac{MN}{P}\left(1 + \frac{\log(P)}{\mu}\right)$ | $\Theta\left(\frac{MN}{P}\log(P)\right)$ | $\Theta\left(\frac{MN}{P}\log(P)\right)$ |
| Repetition [1] | $N\left(1 + \frac{M\log(M)}{P\mu}\right)$ | $\Theta\left(\frac{MN}{P}\log(P)\right)$ | $\Theta\left(N\right)$ |
| MDS | $N\left(1 + \frac{\log(\frac{P}{P-M})}{\mu}\right)$ | $\Theta(N)$ | $\Theta(N)$ |
| Short-Dot | $\frac{N(P-K+M)}{P}\left(1 + \frac{\log(\frac{P}{P-K})}{\mu}\right)$ | $\mathcal{O}(\frac{MN}{P})$ | $\mathcal{O}\left(\frac{MN}{P}\log\left(\frac{P}{M}\right)\right)$ |

[1] A more accurate analysis taking integer effects into account is also presented.

## 1.2 Existing Strategies

**One Single Processor:** For one single processor to compute all $M$ dot-products of length $N$, the computation time is distributed as

$$\Pr(T \le t) = F(t/M) = 1 - \exp\left(-\mu\left(\frac{t}{NM} - 1\right)\right) \quad \forall\, t \ge NM \tag{6}$$

Thus, the expected computation time can be easily derived to be

$$E(T) = MN\left(1 + \frac{1}{\mu}\right) \tag{7}$$

**Uncoded - Divide into $P$ parts and wait for all:** Now, consider an uncoded strategy where the computation is simply divided into $P$ dot-products and sent to $P$ processors. We assume that each processor is sent only one dot-product at a time. We wait for all the processors to finish computation. Note that, integer effects arise when $M$ does not exactly divide $P$. Some rows can be divided among $\lceil \frac{P}{M} \rceil$ processors, while the remaining are divided among $\lfloor \frac{P}{M} \rfloor$ processors. Let $m_1$ and $m_2$ denote the number of rows that get $\lceil \frac{P}{M} \rceil$ processors and $\lfloor \frac{P}{M} \rfloor$ processors respectively. Clearly the values can be obtained by solving:-

$$\begin{bmatrix} 1 & 1 \\ \lceil \frac{P}{M} \rceil & \lfloor \frac{P}{M} \rfloor \end{bmatrix} \begin{bmatrix} m_1 \\ m_2 \end{bmatrix} = \begin{bmatrix} M \\ P \end{bmatrix} \tag{8}$$

Now, we have two groups of exponential variables - one group consisting of $m_1 \lceil \frac{P}{M} \rceil$ independent and identically distributed exponential random variables of task size $\frac{N}{\lceil \frac{P}{M} \rceil}$ and another group consisting of $m_2 \lfloor \frac{P}{M} \rfloor$ independent and identically distributed exponential random variables of task size $\frac{N}{\lfloor \frac{P}{M} \rfloor}$. The two groups are independent of each other. Note that, for each of calculations we assume that $N$ is large compared to $P$ and is divisible by $P, \lfloor \frac{P}{M} \rfloor, \lfloor \frac{P}{M} \rfloor$, so that the integer effects with respect to $N$ do not appear and the plots can be scaled with respect to $N$ for ease of understanding.

The expected computation time is thus given by the expectation of the maximum of all these $P = m_1 \lceil \frac{P}{M} \rceil + m_2 \lfloor \frac{P}{M} \rfloor$ exponential random variables.

$$\Pr(T \le t) = \left( 1 - \exp\left( -\mu \left( \frac{\lceil \frac{P}{M} \rceil t}{N} - 1 \right) \right) \right)^{m_1 \lceil \frac{P}{M} \rceil} \times$$

$$\left( 1 - \exp\left( -\mu \left( \frac{\lfloor \frac{P}{M} \rfloor t}{N} - 1 \right) \right) \right)^{m_2 \lfloor \frac{P}{M} \rfloor} \quad \forall\, t \ge \frac{N}{\lfloor \frac{P}{M} \rfloor} \tag{9}$$

The expectation is thus obtained as

$$E(T) = \int_0^\infty (1 - \Pr(T \le t))\, dt \tag{10}$$

This expression is computed using MATLAB and plotted in the plot of theoretical computation time ( Refer Fig. 1). When $M$ divides $P$ exactly, the expressions are simpler. The computation time for each processor is distributed as

$$\Pr(T \le t) = F(t/M) = 1 - \exp\left( -\mu \left( \frac{Pt}{MN} - 1 \right) \right) \quad \forall\, t \ge NM/P \tag{11}$$

The expected computation time is the maximum of $P$ such independent and identically distributed random variables, as given by:-

$$E(T) = \frac{MN}{P} \left( 1 + \frac{\log(P)}{\mu} \right) \tag{12}$$

The expected time is $\Theta\left( \frac{MN \log(P)}{P} \right)$ whether $M$ is linear or sub-linear in $P$. Our strategy offers a speed-up of $\Omega(\log(P))$ when $M$ is linear in $P$.

**Repetition:** When a $(P, M)$ repetition strategy is used, we separate the matrix into $M$ rows and repeat each row $P/M$ times, so as to obtain a total of $P$ tasks. Note that, integer effects arise when $M$ does not exactly divide $P$. Some rows are repeated $\lceil \frac{P}{M} \rceil$ times, while the remaining are repeated $\lfloor \frac{P}{M} \rfloor$ times. Let $m_1$ and $m_2$ denote the number of rows that are repeated $\lceil \frac{P}{M} \rceil$ times and $\lfloor \frac{P}{M} \rfloor$ times respectively. Clearly the values can be obtained by solving:-

$$\begin{bmatrix} 1 & 1 \\ \lceil \frac{P}{M} \rceil & \lfloor \frac{P}{M} \rfloor \end{bmatrix} \begin{bmatrix} m_1 \\ m_2 \end{bmatrix} = \begin{bmatrix} M \\ P \end{bmatrix} \tag{13}$$

Now, the minimum of $\lceil \frac{P}{M} \rceil$ (or similarly $\lfloor \frac{P}{M} \rfloor$) independent and identically distributed exponential random variables is also exponential with parameter scaled by $\lceil \frac{P}{M} \rceil$ (or similarly $\lfloor \frac{P}{M} \rfloor$). The expected computation time is thus given by the expectation of the maximum of $m_1$ independent exponential variables with parameter scaled by $\lceil \frac{P}{M} \rceil$ and $m_2$ independent exponential variables with parameter scaled by $\lfloor \frac{P}{M} \rfloor$.

$$\Pr(T \leq t) = \left(1 - \exp\left(-\mu \left\lceil \frac{P}{M} \right\rceil \left(\frac{t}{N} - 1\right)\right)\right)^{m_1} \times$$
$$\left(1 - \exp\left(-\mu \left\lfloor \frac{P}{M} \right\rfloor \left(\frac{t}{N} - 1\right)\right)\right)^{m_2} \quad \forall\, t \geq N \quad (14)$$

Figure 2: Theoretical Plot of expected computation time of repetition taking integer effects into account: straggling parameter $\mu = 5$, total processors $P = 1000$ and number of dot-products $M$ is varied from 1 to $P$.

The expectation is thus obtained as

$$E(T) = \int_0^\infty (1 - \Pr(T \leq t))\, dt \quad (15)$$

This expression is computed using MATLAB in the plot of theoretical expected computation time (Fig. 1). When, $M$ exactly divides $P$, the analysis is simpler, and both the two types of exponential distributions are identical. Following an analysis similar to [1], it simplifies to

$$E(T) = N \left(1 + \frac{M \log(M)}{P\mu}\right) \quad (16)$$

When $M$ is linear in $P$, the expected computation time is $\Theta(\frac{MN}{P} \log(P))$ while our strategy achieves $\mathcal{O}(N)$ in this regime. When $M$ is sub-linear in $P$, the expected computation time is $\Theta(N)$ while our strategy Short-Dot achieves $\mathcal{O}\left(\frac{MN \log(P/M)}{P}\right)$ that offers speed-up by a factor diverging to infinity.

**MDS codes-based strategy:** The matrix is separated into $M$ rows and coded into $P$ rows using a $(P, M)$ MDS code. Thus, each processor effectively computes a dot-product of length $N$. We have to wait for any $M$ processors to finish. Assuming the computation of each processor is independent, following an analysis similar to [1], we obtain that,

$$E(T) = N \left(1 + \frac{\log(P)}{\mu} - \frac{\log(P - M)}{\mu}\right) \quad (17)$$

When $M$ is linear in $P$, the expected computation time is $\Theta(N)$ as compared to our strategy that achieves $\mathcal{O}(MN/P)$. However, in the regime where $M$ is sub-linear in $P$, the expected computation time is also $\Theta(N)$ while our strategy achieves $\mathcal{O}\left(\frac{MN \log(P/M)}{P}\right)$, and thus outperforms MDS codes by a factor that diverges to infinity for large $P$.