[Reviews · NeurIPS 2016]

Reviewer 1

Summary

Consider the computation of A.x where A is an M by N matrix and x is an N by 1 vector. Both x and A can be dense. This paper approximates this product by F.x, where F is a P by N matrix, P > M and the row-sparsity of F is N(P-K+M)/P, K is a tunable parameter. The matrix is generated using codes (such as MDS codes, eg Reed-Solomon). This provides a way to speed up computation in parallel systems.

Qualitative Assessment

One of the critical points is that, the main idea of using codes for distributed system speed up seem to be already introduced in references [1],[2],[3],[4]. My sense is that to go beyond the MDS codes structure introduced in those references is the main innovation. Also, relevance to NIPS is a bit limited: it looks more like a distributed systems/performance evaluation paper. This is also clear from the references. This two points make it somewhat below a standard NIPS paper. One minor comment is that the figures are too small and hard to read. On the other hand the experimental section is good, and speeding up distributed systems using codes likely will have applications.

Confidence in this Review

2-Confident (read it all; understood it all reasonably well)


Reviewer 2

Summary

The authors present a sparse linear coding technique, that allows for faster distributed matrix-vector multiplication. The use of additional redundancy resolves the issue of stragglers, that is a well-known bottleneck in distributed architectures. The authors instead of using a dense MDS code, use a sparser code, that allows for faster local computation, and for better delay properties. They conclude with experiments on Condor, that indicate that their approach is promising.

Qualitative Assessment

The paper presents some useful ideas for compensating with delays in distributed computation. From a coding theoretic perspective the novelty is limited, but the application on distributed computation of inner products (matrix-vector) is useful, and relatively novel. My main complaint is on the experimental section. I was hoping to see more comparisons of the proposed codes against MDS and repetition codes. In particular, I am not sure I understand what Figure 6b is showing. It would be useful to see some experiments with respect to k/n that better indicate the points where repetition, or simple MDS codes become better than the presented codes. However -and to be fair- the authors put the effort and run real distributed experiments on Condor, and did not simulate their results using a single machine. A note on Condor: it might be the case that the node delays there are much more substantial compared to a dedicated distributed setup, or even compared to amazon EC2 instances.

Confidence in this Review

3-Expert (read the paper in detail, know the area, quite certain of my opinion)


Reviewer 3

Summary

In this paper, the authors introduce a method to compute a matrix-vector product in parallel. The method relies on a key assumption that the number of available processors is larger than the number of columns in the matrix. Under this assumption, they propose a method that distributes the columns of a modified matrix, computes (sparse) vector dot products in parallel and only has to wait for a small number of the processors to finish computing the dot products. This way, they can avoid the problem of waiting for straggler processors to finish.

Qualitative Assessment

The paper is easy to understand, and the concept seems clear. Detailed comments follow (both major and minor, in no particular order): - My biggest concern about the practicality of the method is the (pretty restrictive) assumption that the number of available processors is larger than the number of columns. Almost all modern applications have a large number of variables to fit, and the number of available processors is almost always smaller (at most in the order of hundreds). So while the math is correct and the results look impressive, i’d like to see this aspect discussed and addressed in detail. - line 10: a only —> only a - Figure 2: at this point, with K > M, short dot is not necessarily faster. it depends on the sparsity (s) as well. You address this later, but this has to be made clear before directing one’s attention to fig 2. - line 142: chosen —> choosing - The flow of ideas to prove theorem 1 is confusing. It is better to separate lemmas from the main theorem proof. - Figure 5 does not paint a full picture. For example, what happens when \mu is varied? Why is MDS worse than no coding at all? Are there any values of \mu for which the repetition scheme is good? Why is the expected time = 1 even when the number of dot products is nearly 0 for MDS and repetition? - line 233: Could you include a plot akin to fig 5 but taking the encoding and decoding complexity also into account? Although it is a one-time computation, it cannot be ignored. - How do the performances of the various algorithms compare on real datasets? You mention linear regression, but experiments on real data would be nice to have.

Confidence in this Review

2-Confident (read it all; understood it all reasonably well)


Reviewer 4

Summary

The authors consider the problem of calculating A^Tx in a distributed setting. In particular, they are concerned with the "straggler effect," in which when work is distributed, some cores take far longer to do the same work, causing substantial delays upon synchronization. The paper introduces Short-Dot, in which redundancy is used to allow the full computation to be completed once only a subset of the processes have finished, before the final stragglers. The authors prove the theoretical backbone to their approach, compare probabilistic run-times of their approach compared with many others, and present experiments demonstrating the practical benefits of their approach.

Qualitative Assessment

The work is interesting, technically sound, correct, and addresses a real problem of interest to many, though not to a degree that I believe would make it worthy of an oral report to the NIPS audience. It is not particularly easy to read - in particular, the presentation of Algorithm 1 without any explanation is quite difficult, even with the setup of Theorem 1. Experiments are very briefly discussed, which is a shame given that I don't know whether the assumptions on the distribution of stragglers are accurate. This would be better served with experiments across different architectures and sizes, since it's such a practical problem (I know, I know, space, but still...). I also worry that this paper may be poorly targeted - this problem is important for fast machine learning, but is probably lower level than the problems most people who follow NIPS are interested in studying directly.

Confidence in this Review

2-Confident (read it all; understood it all reasonably well)


Reviewer 5

Summary

The paper introduces a technique (the “short dot”) to parallelize a single dot product operation across many machines where a small number of machines are likely to be delayed in finishing. This technique can complete the dot product (and move on to the next round of computation) when any K of the P machines complete so it is robust to a small number of stragglers. The “short dot” instead of just using some simple form of replication as previous methods had done, uses some more interesting ideas from coding to get better results — these are small improvements, but are shown optimal for at least a restricted setting. The ideas of this paper are quite neat — I enjoyed it very much. However, very extreme scales are needed where one requires to parallelize a single dot product. As far as I know these settings do occasionally occur, but there are also fundamentally different ways around them (like SGD and sketching). The paper describes one ML example in an iterative method for linear regression — its not clear if it can be extended to any non-linear/kernel settings.

Qualitative Assessment

I think this work will have influence in the parallel computing community. However, I have my doubts of its potential influence on the ML community as it is improves on a scalability problem that few hit, and does not compare against other broad approaches to these very very large scale problems — although this is a lot to ask. To connect better with the ML audience, it would be important to show more detailed connections to other ML applications, and compare against other very very large scale techniques. The combination of ideas and new approaches to coding are very cool. I really enjoyed reading about them. Very well described, written, and illustrated paper.

Confidence in this Review

2-Confident (read it all; understood it all reasonably well)


Reviewer 6

Summary

This paper presents a method to compute large dot products on a distributed system with some stragglers. The authors introduce redundancy in the computation via the proposed error-correction mechanisms, which allows the size of individual dot products computed at each processor to be shorter than the length of the input.

Qualitative Assessment

This work is a sparse extension to the existing MDS coding method. My concerns are as follows. 1. Does the number of stragglers affect the performance of Short-Dot method? In experiments, if we have known which processors are stragglers, does the proposed Short-Dot method still outperform the existing methods on the healthy processors? 2. The proposed method is based on the assumption that any K columns of the matrix F can be sufficient to generate the original matrix A (N\times M), the experiments demonstrate its performance when M is small (M is 10), what will happen when M is large, after all, most of data are large-scale in real applications? 3. In experiments, how to set K and P according to M? 4. Typo: Table 2 in supplement, the 1st processor column and rows of K=30 and K=40, 08.12 sec->8.12 sec, 04.02 sec->4.02 sec.

Confidence in this Review

2-Confident (read it all; understood it all reasonably well)